# Short-term costs and cost-efficiency of HPV triage strategies in a high HIV-prevalence setting: Evidence from Botswana

Adriane Wynn[1,2]*, Devon Harris[3], Anna Modest[3], Maria de Fatima Reyes[3], Bridgette Wamakima[4], Kelebogile Gaborone[2], Natasha Moraka[2], Annika Gompers[3], Sikhulile Moyo[2,5,6], Roger Shapiro[5], Doreen Ramogola-Masire[7], Rebecca Luckett[3,2,7]

1 Division of Infectious Diseases and Global Public Health, University of California San Diego, La Jolla, California, United States of America, 2 Botswana Harvard Health Partnership, Gaborone, Botswana, 3 Department of Obstetrics and Gynecology, Beth Israel Deaconess Medical Center, Boston, Massachusetts, United States of America, 4 Department of Obstetrics and Gynecology at Brigham and Women's Hospital, Boston, United States of America, 5 Department of Immunology and Infectious Diseases, Harvard T.H. Chan School of Public Health, Boston, United States of America, 6 Department of Pathology, Division of Medical Virology, Stellenbosch University, Cape Town, South Africa, 7 The Department of Obstetrics & Gynaecology, University of Botswana, Gaborone, Botswana

* awynn@health.ucsd.edu

## Abstract

Cervical cancer remains the leading cause of cancer death among women in sub-Saharan Africa and is more severe in high HIV-burdened countries due to persistent high-risk human papillomavirus (hrHPV). In 2021, the World Health Organization recommended primary hrHPV testing for cervical cancer screening; however, optimal triage strategies following positive hrHPV tests remain unclear. We conducted a prospective cost analysis of triage methods for positive hrHPV results among women living with and without HIV in Gaborone, Botswana. We used a micro-costing approach from the perspective of the healthcare provider. The main outcomes were the implementation costs associated with three triage strategies following hrHPV testing: 8-type HPV genotype restriction, visual inspection with acetic acid (VIA), and colposcopy. We also compared the strategies by measuring the change in costs divided by the change in number of true cases of cervical intraepithelial neoplasia (CIN) 2 or worse (CIN2+) identified, based on the results of a prospective cohort study. Results indicated that the 8-type HPV genotype restriction strategy was the most cost-efficient, requiring no additional costs beyond hrHPV testing and identifying the highest number of true CIN2+cases. VIA and colposcopy triage identified fewer true cases of CIN2+ and incurred additional costs, with colposcopy being the most expensive. Results were consistent in women with and without HIV. Sensitivity analysis highlighted personnel and hrHPV test kit cartridge costs as significant drivers of overall costs. Post-hoc analysis incorporating average treatment costs for precancer demonstrated that genotyping remained dominant at lower treatment costs but

**Data availability statement:** All data relevant to the costing study are within the manuscript and its Supporting Information files (minimal dataset). For the parent study (that assessed the specificity and sensitivity of the triage strategies) the datasets are available from the corresponding author on reasonable request.

**Funding:** This work was supported by the National Institutes of Health (National Institute of Alcohol Abuse and Alcoholism K01AA027733 to AW; National Cancer Institute CASCADE HIV/Cervical Cancer Prevention Clinical Trials Network U24CA275417 to AW and RL, and K08CA271949 to RL; Fogarty International Center K43 TW012350-01 to SM). The funders did not play a role in the study design, data collection and analysis, decision to publish, or preparation of the manuscript.

**Competing interests:** The authors have declared that no competing interests exist.

became less favorable as treatment costs increased. We found that 8-type genotype restriction was optimal compared to hrHPV screening combined with VIA or colposcopy. Cost estimates can inform future studies that examine the long-term costs and health outcomes of HPV-based two-stage screening algorithms.

## Introduction

Cervical cancer remains the leading cause of cancer death among women in sub-Saharan Africa despite it being preventable through prophylactic vaccination, screening, and treatment of dysplasia [1]. Cervical cancer is particularly devastating in countries with a high burden of the human immunodeficiency virus (HIV), occurring more frequently and at earlier ages due to high rates of persistent high-risk human papillomavirus (hrHPV), the causative agent of most cervical cancer [2–4]. The inequity in the burden of cervical cancer in this region is a direct result of both lack of effective and accessible screening modalities and the high prevalence of HIV. Botswana has the third highest HIV prevalence in the world [5]. Recent data demonstrated a prevalence of hrHPV of 56% amongst a cohort of women living with HIV (WLHIV) in Botswana [6]. This appears consistent with data from various other African countries which demonstrate prevalence rates of 30%−50% among WLHIV. Data regarding the prevalence of high-grade dysplasia in WLHIV however, is highly variable with rates of 10–54% [7–10].

In 2021, the World Health Organization (WHO) guidelines for cervical cancer screening emphasized the importance of a global move toward high-performance primary hrHPV testing in order to achieve the elimination of cervical cancer [11] hrHPV testing has the potential for rapid scale-up and increased access to screening through self-sampling, making it a highly attractive option for screening in sub-Saharan Africa where screening coverage has lagged [12–14]. While hrHPV testing has the highest sensitivity for detecting cervical dysplasia and is the most effective primary screening strategy available [15,16], the positive predictive value for cervical dysplasia is low, necessitating additional triage [17,18]. Currently, the ideal triage strategy has not been determined, though restrictive genotyping, visual inspection with acetic acid (VIA), and colposcopy all remain potential strategies, particularly where cytology is not available at a population level. For example, in Botswana, cytology is not deemed logistically feasible by the National Cervical Cancer Prevention Program as a triage strategy due to cost and personnel requirements [19]. Further, previous research showed that cytology had equivalent performance compared to VIA [6]. The costs associated with these two-step triage strategies is a critical consideration in supporting sustainable, effective cervical screening programmes in low-and-middle income countries (LMICs), particularly those with a high prevalence of HIV and corresponding high rates of hrHPV positivity.

While prior studies have demonstrated that primary hrHPV screening followed directly by treatment is a cost-effective strategy, many of these analyses were performed in lower risk areas, influencing their results and limiting their generalizability

to countries like Botswana [20–23]. Additionally, current WHO guidelines recommend triage of positive non-16/18 hrHPV results prior to treatment in WLHIV to avoid overtreatment [11]. Our objective was thus to expand on the available data regarding cost to include an evaluation of the available two-step triage strategies in our context.

Our study was conducted among women living with and without HIV in Botswana and estimates the average per-screening costs of three two-stage screening algorithms that begin with primary hrHPV testing followed by 1) 8-type HPV genotype restriction (16/18/31/33/35/45/52/58), 2) VIA or 3) colposcopy for those who test positive for hrHPV. We also compared the strategies by estimating the incremental costs per case of cervical intraepithelial neoplasia (CIN) 2 or worse (CIN2+) diagnosed.

## Methods

### Study design

We conducted a prospective cost analysis of triage methods for positive hrHPV results among women living with and without HIV from the perspective of the health care provider. The main outcomes were the implementation costs of the primary hrHPV screen and the additional cost of three triage strategies among woman who tested hrHPV positive. We also compared the three triage strategies using incremental cost-effectiveness ratios, which measured the change in costs divided by the change in the number of true cases of CIN2+ identified, based on the results of a prospective cohort study in Botswana [7]. Results are reported in accordance with the Consolidated Health Economic Evaluation Reporting Standards (CHEERS) [24].

### Study population and setting

A prospective cohort study enrolled 3,000 women to evaluate the performance of primary hrHPV DNA testing with genotyping available followed by triage for those who tested positive for the 8 most oncogenic HPV types (16/18/31/33/35/45/52/58, referred to as "8-type HPV genotype restriction"), VIA, and colposcopy in South East District, Botswana. Recruitment, including a study flow diagram implementation, and outcomes (e.g., sensitivities and specificities of triage strategies) have been previously described [7]. In brief, women were recruited from health facilities in the South East District, Botswana between February 2021 and July 2022 with follow-up visits completed by February 2023. Participation was open to women visiting district health facilities for any service, thus representing the general screening population. Eligibility criteria were women, ≥ 25 years of age, not pregnant, with an intact cervix, no prior cervical cancer diagnosis, and competent to understand study procedures and give informed consent. Written informed consent was obtained, and those who enrolled responded to a questionnaire that collected data on socio-demographics and health history, including HIV status. For the women presumed to be HIV uninfected, evidence of a negative HIV test within the past year was requested. If no recent HIV test, participants were referred for HIV testing (offered on-site) prior to examination. Any new HIV diagnoses were referred for baseline laboratory testing and treatment per Botswana national guidelines.

Among the 2,969 women who underwent primary hrHPV testing, 1,269 tested positive for high-risk HPV, underwent triage evaluation and had histopathology results available. This included 688 women living with HIV and 581 without.

### HPV screening and triage strategies

After a brief explanation, participants self-collected HPV vaginal samples. All samples were transported to the Botswana Harvard Partnership Reference Laboratory in Gaborone for testing. HPV specimens were tested for 15 hrHPV types (16,18,31,33,35,39,45,51,52,53,56,58,59,66,68) using the AmpFire® HPV Assay (Atila BioSystems, Mountain View, California), on a high through-put realtime-PCR platform that is already established in Botswana.

The first triage strategy was management of women positive for only the 8 HPV genotypes most associated with cervical cancer (HPV 16,18,31,33,35,45,52,58). For this strategy, we assumed that women would be referred for immediate

treatment. For the VIA and colposcopy triage strategies, participants who tested positive for hrHPV were contacted and asked to return for additional screening and biopsy. At the VIA and colposcopy triage visit, participants underwent a speculum examination where 5% acetic acid was applied to the cervix using a cotton swab. Visual inspection with acetic acid (VIA) was performed by a nurse midwife with Botswana Ministry of Health and Wellness national VIA training and experience performing VIA in a clinical setting. The VIA nurse recorded their impression. VIA findings were categorized as normal, lesion eligible for ablation (low-grade impression), or lesion requiring referral for LEEP (high-grade impression). Subsequently, a gynecologist blinded to the VIA assessment performed colposcopy and recorded their impression. Colposcopy impression was recorded as normal, low-grade and high-grade. All participants undergoing triage testing also had a biopsy collected at the time of colposcopy. If there was a visible lesion, a punch biopsy or LEEP was performed according to current best practice in Botswana. If no lesion was visible, a small endocervical excision or an endocervical curettage was performed. Pathology specimens were processed by the Botswana National Health Laboratory, and results were reported according to the cervical intraepithelial neoplasia (CIN) classification system [25] with categorization by severity. Participants with CIN2+ on biopsy or endocervical curettage were referred for an excisional procedure if not already performed. Women with histopathology showing CIN3 with microinvasion, adenocarcinoma-in-situ or invasive cervical cancer were referred to a multidisciplinary gynaecologic oncology clinic for further assessment and treatment. Participants who tested negative for hrHPV were counseled to repeat screening in three- or five-year intervals, depending on whether they were living with or without HIV, respectively.

### Identification and calculation of costs

Cost data were collected prospectively between September 2021 and August 2022 using a micro-costing approach, [26] where each component of health care utilized was recorded and a unit cost was applied to each component. Costs were collected in the original currency of purchase, inflated using the World Bank GDP deflator, and converted to 2022 United States dollars using the World Bank official exchange rate.

To determine personnel utilization and direct labor costs, time-and-motion observations were conducted by a member of the study team. We observed six participants undergoing hrHPV testing, nine undergoing VIA screening, and 20 participants undergoing colposcopy. The number of observations was determined both by staff capacity and complexity of the procedure. Results were discussed with experts (providers who routinely deliver cervical cancer screening in Botswana) to confirm the face validity of time spent during usual practice. Average staff time required to process the hrHPV samples was reported by the Botswana Harvard Partnership Reference Laboratory. Botswana Ministry of Health and Wellness Staff salaries were obtained from the Botswana Directorate of Public Service Management's basic salary scale, which included basic salary and housing allowance. We divided the annual salary by the estimated number of working days in Botswana (220) and assumed a 40-hour work week. The amount of time spent on each task by each worker was multiplied by the rate per minute of each employee paid to perform the task. Care was taken to estimate costs for activities directly related to the procedures, and costs related to the study were excluded.

The use and number or proportion of supplies and capital utilized were also recorded while staff were conducting the time-and-motion observations. Supply and capital costs were obtained from the Botswana Central Medical Stores availability report and study procurement invoices. The hrHPV cartridge and sample collection kit were based on negotiated prices if procured by the Government. The cost of supplies was calculated by multiplying the quantity used per each activity by its cost. Capital costs were annualized based on their use life and a discount rate of 3% and divided by annual throughputs. As multiple PCR platforms are in use in Botswana, we estimated a per cartridge lab charge, which is reflected in personnel costs.

Training costs were estimated based on a 10-day program for doctors and nurses that included room and board, per diem, training staff salaries, and the salaries of providers being trained. We assumed that training occurred every five years. Overhead costs were not included.

## Health outcomes

The health outcomes have been previously reported [7] and include the joint sensitivities and specificities of the two-stage screening strategy. In brief, among the 2,969 women screened for HPV, 1,480 were positive and 1,269 had histopathology data for analysis. The 8-type genotype restriction strategy had the highest sensitivity to detect CIN2+ (88% of WLHIV, 86% without HIV), followed by colposcopy (71% WLHIV, 46% without HIV), and VIA (62% WLHIV+, 44% without HIV). However, specificity was low (30% of WLHIV, 37% without HIV) compared to VIA (72% of WLHIV, 75% without HIV) and colposcopy (66% of WLHIV, 71% without HIV). The health outcome is the number of histologically confirmed cases of CIN2+.

## Analysis

Data were entered into a Microsoft Excel database. The time frame included the initial screening test followed by a triage screen. First, we calculated the unit cost for the primary screen and triage screens and estimated the component costs (e.g., capital, supplies, personnel) for each. To estimate the total screening costs, we multiplied the cost of the primary screen by the number screened and added the cost of the triage strategies multiplied by the number of women who screened positive on the primary screen. Next, we compared the costs and true cases of CIN2+ identified for each triage strategy. Strategies were ranked according to cost with the lowest cost strategy as the base case, dominated strategies were removed, and the remaining strategies were compared using an incremental cost-effectiveness ratios (ICER) using the following formula:

$$ICER = Cost_{strategy2} - Cost_{strategy1} / Cases_{strategy2} - Cases_{strategy1}.$$

## Sensitivity analyses

Parameters were varied in one-way sensitivity analysis to assess the impact on the unit costs of each strategy. High and low values for personnel were based on time-and-motion observations and supplies and capital costs were varied by +/- 50%. We also estimated best and worst-case scenarios that included the lowest and highest values for the cost parameters. Further, we compared the characteristics of participants with and without pathology, stratified by HIV status.

## Post-hoc analysis

The primary goal of this study was to estimate the costs associated with three HPV triage strategies. As a post-hoc analysis, we incorporated treatment costs among both true and false positive cases of precancer in each strategy. Treatment costs were based on cost estimates of loop electrosurgical excision procedure (LEEP) derived from our study as well as treatment costs found in the literature [27]. We then calculated ICERs and displayed the results in a cost-effectiveness planes.

## Ethics

The institutional review boards of the Botswana Ministry of Health and Wellness (13/18/1), the University of Botswana (URB/IRB/1543), Beth Israel Deaconess Medical Center (2019P001130) and the South East District Health Management Team approved the parent study.

## Results

Table 1 describes first and second stage screening strategies, including the cost components for each. The majority of activities (e.g., counselling, instructions, examination, paperwork, and patient results) were carried out by a mid-level

**Table 1. Summary of three high-risk (hr)HPV triage strategies for detecting CIN2+ among women in South East District, Botswana.**

| Triage Strategy | Description | Cost Components | | | |
|---|---|---|---|---|---|
| | | Training | Capital | Supplies | Personnel |
| **Primary screening** | | | | | |
| hrHPV screening | All screened for high-risk HPV. If positive, referred for triage test. | All staff participate in the MoHW cervical cancer prevention screening and treatment training and refresh every five years. | Utilize existing PCR testing platforms distributed throughout the country | Self-collection kit, hrHPV cartridge/assay, transport charge, shipping & customs | Nurse time to explain specimen collection, counsel, label & store, provide results, paperwork. Lab tech time to process sample and provide results to clinic. |
| **Triage strategy** | | | | | |
| 1 **8-type HPV genotype restriction (included in the primary HPV screen)** | Participants who are 8-type HPV genotype positive are referred for treatment. | | No additional costs beyond first-stage hrHPV screening | | |
| 2 **VIA** | Participants hrHPV+ asked to return for a 2nd visit. Visual inspection performed and those with suspicion of cervical precancer are treated. | | Speculums, sponge holding forceps, medicine trolley, examination light | Acetic acid, distilled water, cotton wool, disinfectants, urine collection cup, pregnancy test strip, water soluble lubricant | Nurse time for counseling, VIA exam, counsel on findings & follow-up, paperwork |
| 3 **Colposcopy** | Participants hrHPV+ asked to return for a 2nd visit. Colposcopy performed and those with suspicion of cervical precancer are treated. | | Speculums, sponge holding forceps, medicine trolley, examination light, colposcope | Acetic acid, distilled water, cotton wool, disinfectants, urine collection cup, pregnancy test strip, water soluble lubricant | Doctor time for counseling, colposcopic exam, counsel on findings & follow-up, paperwork |

Notes: hrHPV = High risk human papillomavirus, MoHW = Ministry of Health and Wellness, VIA=Visual inspection with acetic acid, CIN = high-grade cervical intraepithelial neoplasia. Costs of treatment provided or prevented are not considered in this analysis.

8-type HPV genotype restriction includes HPV types 16/18/31/33/35/45/52/58.

nurse. The colposcopy counseling, exam, and paperwork were conducted by a doctor. Supplies and equipment were similar between VIA and colposcopy; however, colposcopy required additional equipment (e.g., colposcope). We assumed that triage with 8-type HPV genotype restriction required no additional personnel, supplies, or equipment. Our table of parameters can be found in Supplementary Information Table 1. Table 2 shows the estimated individual base test costs and components for HPV testing and genotyping, VIA and colposcopy. For hrHPV, time-and-motion observations identified 16 minutes as the average time for provider counseling, instruction for specimen collection, paperwork, and results. The Botswana Harvard Partnership Reference Laboratory reported that sample processing took an average of seven minutes and no capital costs were included. Personnel time was observed to be similar between the VIA and colposcopy (with different personnel delivering each service).

Outcomes were based on 2,957 women who underwent a primary hrHPV DNA test,1,293 women screened positive for high-risk HPV, completed triage, and had histopathology for evaluation; and 206 confirmed cases of CIN2+ [7]. **Table 3** indicates the costs and outcomes of screening this population using the two-stage triage strategies, stratified by HIV status. Among all 1,269 participants undergoing triage, the primary HPV test cost was $52,442, genotype restriction was included in the primary screen and incurred no additional cost, VIA and colposcopy incurred an additional $5,400 and $15,203 respectively. With the lowest additional cost and largest number of true CIN2+cases identified the 8-type HPV genotype restriction was the dominant strategy (less costly and more effective) among the entire sample and stratified by HIV status.

**Table 2. Clinical and laboratory time, supplies, and capital costs per person screened (2022 US Dollars).**

| Procedure | Mean clinical time (minutes) | Mean laboratory time (minutes) | Training costs | Personnel Costs | Supplies/ Recurrent Costs | Capital Costs | Total Cost |
|---|---|---|---|---|---|---|---|
| *Primary screening* | | | | | | | |
| HPV | 16 (8-22) | 7 (6-9) | $1.56 ($0-7.80) | $3.01 ($1.80-22) | $13.16 ($5.48-19.50) | $0 | $17.72 ($7.28-49.30) |
| *Triage screening* | | | | | | | |
| Genotyping (included in primary HPV screen) | 0 | | | $0 | $0 | $0 | No additional cost |
| VIA | 25 (19-44) | | | $2.82 ($1.25-4.65) | $1.07 ($0.53-1.60) | $0.37 ($0.29-0.44) | $4.26 ($2.07-$6.69) |
| Colposcopy | 25 (16-32) | | | $8.92 ($5.60-13.44) | $1.07 ($0.53-1.60) | $1.99 ($1.60-2.39) | $11.98 ($7.73-$17.43) |

Notes: High and low values for personnel were based on time-and-motion observations and supplies and capital costs were varied by +/- 50%.

**Table 3. Costs, effectiveness, and incremental cost-effectiveness of true cases of CIN2 + detected among a cohort of 2,959 participants who received primary hrHPV screening followed by three triage strategies among 1,269 participants who screened positive for high-risk HPV in Botswana.**

| Test Option Stage 1: hrHPV + | Cost of primary hrHPV screen | Additional Cost of Triage Screen | True CIN2 + Cases Detected (95% Confidence intervals) | Average Cost Per True Case (including primary and triage screening costs) | Incremental Cost Effectiveness Ratio |
|---|---|---|---|---|---|
| *Women with HIV* | | | | | |
| Genotyping† | $26,194 | $0 | 115 (106-122) | $228 (214-247) | Dominant strategy* |
| VIA | | $2,928 | 81 (69-92) | $360 (285-380) | |
| Colposcopy | | $8,243 | 92 (81-101) | $374 (259-323) | |
| | | | | | |
| *Women without HIV* | | | | | |
| Genotyping† | $26,247 | $0 | 61 (54-66) | $430 (398-486) | Dominant strategy* |
| VIA | | $2,472 | 31 (12-40) | $926 (656−2,187) | |
| Colposcopy | | $6,961 | 32 (24-41) | $1038 (640−1,093) | |
| | | | | | |
| *All* | | | | | |
| Genotyping† | $52,442 | $0 | 176 (160-188) | $298 (278-328) | Dominant strategy* |
| VIA | | $5,400 | 112 (81-132) | $516 (397-647) | |
| Colposcopy | | $15,203 | 124 (105-142) | $546 (369-499) | |

Notes: †Genotyping was included in the primary HPV screening test. *Dominant strategy = less costly and more effective; hrHPV = High risk human papillomavirus, VIA = Visual inspection with acetic acid, CIN = high-grade cervical intraepithelial neoplasia. The incremental cost effectiveness ratio is based on the change in cost divided by the change in CIN2 + cases averted.

The sensitivity analysis (Fig 1). found that the per patient personnel and hrHPV test kit cartridge costs were the biggest cost drivers. When setting parameters to their highest and lowest values, the per-person hrHPV and genotyping costs ranged from $9 to $50. For VIA and colposcopy triage, personnel followed by supplies were the biggest cost drivers. When setting parameters to high and low values, VIA ranged from $2-$7 and colposcopy ranged from $9-$17 per person, in addition to the hrHPV DNA testing (with genotyping) cost. (See S1 Table in S1 File for the base case and high and low estimates used in the sensitivity analysis). In our analysis that compared the characteristics of participants with and without pathology, we found that women living with HIV with pathology had slightly longer durations of living with HIV and ART

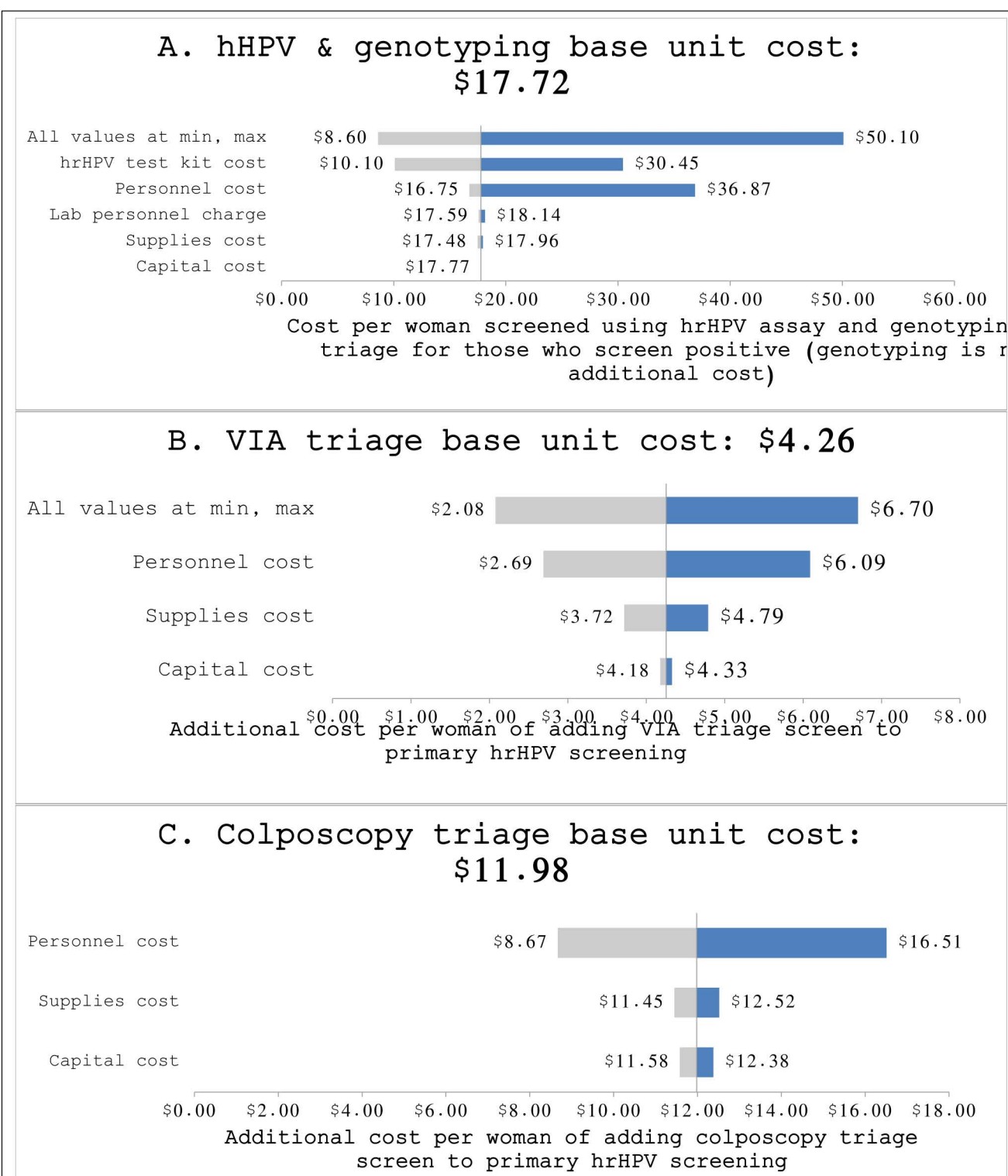

**Fig 1. One-way sensitivity analysis. (A)** Primary hrHPV screen, which includes the genotyping triage strategy for those who screened positive. Genotyping has no additional cost. **(B)** The additional cost of adding VIA as a triage screen for those who screened positive using the hrHPV assay. **(C)** The additional cost of adding colposcopy as a triage screen for those who screened positive using the hrHPV assay.

use, compared to those without pathology. Among participants with and without HIV, participants with pathology were more likely to self-report prior history of cervical screening.

In our post-hoc analysis of short-term treatment costs, we applied the cost per person treated with LEEP that we estimated during our study ($16) as well as higher costs found in the literature ($32 and $64). When we included treatment, genotyping no longer had strong dominance over VIA and colposcopy (Fig 2.), because of its far higher sensitivity and lower specificity in detecting CIN2 + . At a treatment cost of $16 per person, genotyping would have an incremental cost of $39 per true case of CIN2 + identified, compared to VIA. At a treatment cost of $32 per person, genotyping would have an incremental cost of $163 per true case of CIN2 + identified, compared to VIA. At a treatment cost of $64 per person, genotyping was more expensive than VIA and colposcopy. Compared to VIA, colposcopy would have an incremental cost of $1,137 per true case of CIN2 + identified. Compared to colposcopy, genotyping would have an incremental cost of $242 per true case of CIN2 + identified. We found that genotyping became costlier than VIA when treatment costs exceeded $10 and costlier than colposcopy when they exceeded $36. (S1 Fig in S1 File). A table of the outcomes, costs, and ICERS can be found in the supplement (S2 Table in S1 File). These program costs for treating true cases of CIN2 + do not account for the health benefits of treating these lesions, or any long-term cost reduction from reduced cervical cancer burden in the future.

## Discussion

We estimated the costs associated with three hrHPV triage strategies among a mixed cohort of women with and without HIV from the perspective of the healthcare provider in Botswana. As all strategies included the cost of hrHPV DNA testing, the 8-type HPV genotype restriction triage strategy was the lowest because it was included in the primary screen and added no additional costs. Thereafter, hrHPV with VIA triage had the second lowest unit cost followed by hrHPV with colposcopy triage. Additionally, we compared the incremental costs of each strategy to the number of true CIN2 + cases identified among women who screened positive. We found that triage with 8-type HPV genotype restriction was the dominant strategy with a lower cost and higher number of true cases detected, compared to VIA and colposcopy. A post-hoc analysis found that because of the additional true and false positive cases identified by 8-type HPV genotype restriction, it became the most expensive triage strategy when treatment costs increased.

Our cost estimations were similar to previous studies in Sub-Saharan Africa. A recent study in Burkina Faso found that per-person screening costs were $3.20 for VIA and $6.60 for colposcopy; however, the point-of-care hrHPV strategy using careHPV ($27.30 per person) was higher than our estimate because authors included the capital costs of the machine [28]. Our costs were lower than a study that assessed the costs of integrating cervical screening (using the careHPV test) into HIV care in Kenya [29]. The authors found that the marginal cost of standalone cervical screening using VIA was $17.54 and a rapid hrHPV DNA test was $32.51. However, this study included patient costs (e.g., loss of earnings, transport, and child/elderly care) and overhead. Our VIA estimates were similar to a 2015 study in South Africa ($3.67); however, our hrHPV testing cost estimates were lower than the cost in the same study in South Africa ($54.34), which included a $51.74 lab and transport fee.

Many studies have examined the cost-effectiveness of primary screening strategies, but fewer have assessed triage following primary hrHPV DNA testing. The study in Burkina Faso found that compared to VIA alone, the incremental cost of careHPV per CIN2 + case detected was $814; and adding careHPV as a triage test to a primary VIA screen was not cost-effective [28]. The WHO Guidelines Development Group modelled the benefits, harms, and costs of several cervical cancer primary and triage screening strategies in 78 LMICs [30]. This research found that primary hrHPV screening followed by HPV 16/18 restriction, VIA, or colposcopy had similar results to primary HPV DNA alone, in terms of the incremental cost per healthy adjusted life-year saved. This result was due to the added cost of the triage test and, while triage increased specificity (and reduced unnecessary treatment), the triage test decreased the sensitivity of hrHPV DNA testing alone, which thus reduced the overall effectiveness of the HPV-based screening strategy. However, authors stated

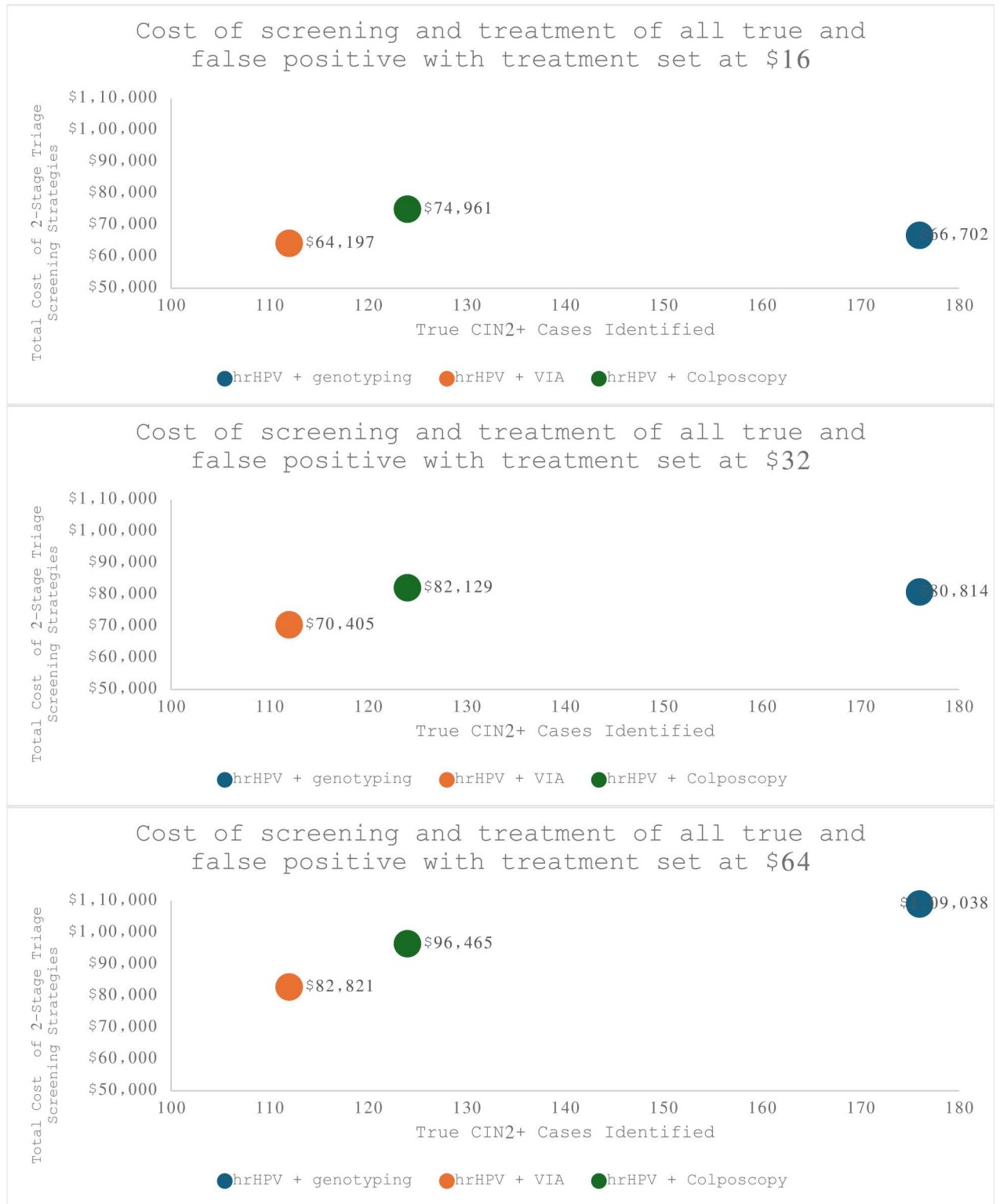

**Fig 2. Post-hoc analysis examining the cost-effectiveness plane of three HPV triage strategies including varied short-term treatment for true and false positive CIN2+.**

that results are highly dependent on follow-up and treatment assumptions and that decisions regarding triaging strategies must be contextualized to the setting and country.

A major obstacle to the adoption of hrHPV screening globally has been the cost of the test itself, compared to the cost of VIA. VIA has consistently been shown to have low sensitivity as a primary screening and triage test [6,7]. Our data demonstrate the additional cost of triage with VIA compared to 8-type HPV genotype restriction; however, these results will not be generalizable to hrHPV triage strategies where genotyping is not provided alongside the primary HPV result. In this cost analysis, we did not account for the future costs of true cases missed by the less effective strategies, including treatment of persistent cervical dysplasia and progression to cervical cancer, which, although discounted, would likely surpass the short-term costs of treatment we included in our post-hoc analysis. Our data can inform additional analyses that incorporate future costs and health outcomes. Finally, our data highlight the need to negotiate lower prices for HPV tests as they are driving the cost of primary hrHPV screening algorithms.

While our study included the costs of primary hrHPV screening and three strategies for triaging women who were hrHPV positive, we did not collect costs associated with the subsequent treatment and rescreening cascade. Compared to VIA and colposcopy triage, hrHPV genotyping more accurately narrows the pool of people who need further evaluation; however, the impact of this strategy is dependent on the subsequent diagnostic and/or treatment procedures. For example, if the diagnostic procedure is VIA, many true cases will still be missed. VIA and colposcopy have limitations, particularly in detecting endocervical lesions. Women with HIV may have increased inflammation and non-HPV-related cervicitis, making visible lesions more common and potentially larger, leading to higher VIA positivity rates [31–33]. Further, if the next step is immediate treatment (e.g., LEEP or thermal ablation), overtreatment will occur. This weakness is partially demonstrated by our post-hoc analysis that found hrHPV genotyping became costlier than VIA and colposcopy at relatively low treatment costs due to higher sensitivity and lower specificity. In long-term mathematical modeling studies, preventing the costs of cancer care can balance out the costs of overtreatment [30]. However, the harms caused by overtreatment are not well measured, there is no widely accepted threshold on the maximum number of cases of overtreatment that are acceptable in order to prevent an additional case of cervical cancer, and more research is needed [34]. In practice, the next steps following an 8-type hrHPV genotype restriction would vary by context. For example, higher-resource settings may opt for routine biopsies to determine treatment needs, while others may rely on VIA. These findings highlight the need for tailored triage strategies that consider both resource availability and patient-specific factors such as HIV status.

Our study has several strengths, including that cost data were collected prospectively alongside a large validation study with a valid histological outcome. However, our study has limitations. We did not calculate the patient costs associated with each strategy, which may have included transport, lost wages and childcare needs required to attend additional visits. To reduce patient costs, all strategies could benefit from providing same-day primary results, triage, and treatment. Prior to implementation and scale-up of primary HPV screening followed by triage, it is important to consider patient preferences and potential patient costs which would likely impact adherence to screening completion. Also, the lab personnel time estimates for processing hrHPV tests are based on average calculations, which do not account for staff idle time. However, the sensitivity analysis included a lower bound on the average number of specimens processed per day, which may partially capture the impact of downtime on costs. The cost estimates did not include the capital costs of the PCR platform used for the primary hrHPV test and genotype restriction, which reflects the benefit of integrating with existing laboratory capacity where possible. If such a strategy could be brought to scale, cervical cancer screening would need to be factored into the overall country laboratory planning. Loss to follow-up occurred among 14% of participants, which resulted in missing pathology results. In our sensitivity analysis, we found limited differences between participants who were and were not missing data. While participants missing pathology were more likely to have a shorter duration of living with HIV and being on ART, previous research has not found a significant difference between HIV and ART duration and increased risk for CIN 2+ [35]. However, participants with pathology were more likely to self-report prior receipt of cervical

cancer screenings than those missing pathology. Thus, if our sample included women less likely to have CIN2+ (because they were more likely to have been screened and treated), the average cost per CIN2+detected would be biased higher than what would be found in the general population for all strategies. Additionally, we did not include costs associated with biopsy based on triage results nor the costs of subsequent management (i.e., biopsy vs ablation vs other strategy). Biopsy is not standard for VIA-based triage strategies, and while it may reduce overtreatment, it would introduce a third visit to complete treatment as needed. As mentioned above, we did not consider the long-term costs of cancer care for patients who would have been missed by the VIA and colposcopy strategies.

We estimated the costs associated with three hrHPV triage strategies among a cohort of women in Botswana. We also examined the costs per true case of CIN2+identified. Our results found that 8-type genotype restriction had the lowest cost and identified the most true cases of CIN2+, compared to hrHPV screening combined with VIA or colposcopy, regardless of HIV status. Our analysis also underscores the need for lower prices for HPV tests and our cost estimates can inform future studies that model the lifetime costs and benefits of HPV-based two-stage screening algorithms.

## Supporting information

**S1 File.** This file contains: S1 Table. Cost parameters collected through micro-costing (2022 USD). S1 Figure: Post-hoc analysis of total costs of hrHP screening followed by three triage strategies, including screening and treatment costs, by modelled increasing treatment costs. S2 Table. Post-hoc analysis with varied short-term costs of treatment of both true and false positive cases of CIN2+detected using three triage strategies following HPV positivity. Costs do not reflect long term treatment or delayed treatment and health outcomes for true cases missed. S3a: Demographic characteristics for those with and without pathology, among participants living with HIV. S3b: Demographic characteristics for those with and without pathology, among women without HIV.
(DOCX)

## Author contributions

**Conceptualization:** Adriane Wynn, Roger Shapiro, Doreen Ramogola-Masire, Rebecca Luckett.

**Data curation:** Anna Modest, Natasha Moraka, Rebecca Luckett.

**Formal analysis:** Adriane Wynn, Anna Modest, Kelebogile Gaborone.

**Investigation:** Kelebogile Gaborone.

**Methodology:** Adriane Wynn, Anna Modest, Annika Gompers, Sikhulile Moyo, Doreen Ramogola-Masire, Rebecca Luckett.

**Validation:** Devon Harris, Maria de Fatima Reyes, Bridgette Wamakima.

**Writing – original draft:** Adriane Wynn, Bridgette Wamakima, Annika Gompers, Sikhulile Moyo, Rebecca Luckett.

**Writing – review & editing:** Devon Harris, Anna Modest, Maria de Fatima Reyes, Kelebogile Gaborone, Natasha Moraka, Roger Shapiro, Doreen Ramogola-Masire.

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
