## [Decision Letter · Decision Letter 0]

29 Nov 2024

Dear Dr. Wynn,

We look forward to receiving your revised manuscript.

Kind regards,

Ivan Sabol

Academic Editor

PLOS ONE

Journal Requirements:

3. We note that you have indicated that there are restrictions to data sharing for this study. PLOS only allows data to be available upon request if there are legal or ethical restrictions on sharing data publicly. For more information on unacceptable data access restrictions, please see http://journals.plos.org/plosone/s/data-availability#loc-unacceptable-data-access-restrictions. Before we proceed with your manuscript, please address the following prompts: a) If there are ethical or legal restrictions on sharing a de-identified data set, please explain them in detail (e.g., data contain potentially identifying or sensitive patient information, data are owned by a third-party organization, etc.) and who has imposed them (e.g., a Research Ethics Committee or Institutional Review Board, etc.). Please also provide contact information for a data access committee, ethics committee, or other institutional body to which data requests may be sent. b) If there are no restrictions, please upload the minimal anonymized data set necessary to replicate your study findings to a stable, public repository and provide us with the relevant URLs, DOIs, or accession numbers. For a list of recommended repositories, please see https://journals.plos.org/plosone/s/recommended-repositories. You also have the option of uploading the data as Supporting Information files, but we would recommend depositing data directly to a data repository if possible. We will update your Data Availability statement on your behalf to reflect the information you provide.

Additional Editor Comments:

P12 “and treatment of dysplasia.5,1“ references not sequential

P12 “While hrHPV testing has the highest sensitivity and specificity for detecting cervical dysplasia“ it might be misleading to state hrHPV testing has the highest specificity especially as ref17 acknowledges that cytology had higher specificity. Ref 16 might be outdated (2006)

P13 “A prospective cohort study enrolled 3,000 women to evaluate the performance of primary hrHPV DNA testing followed by triage…“ it might be beneficial to specify the reasons why the patients originaly visited the health facilities? General screening population or some higher risk population (almost 50% were hrHPV positive)?

P14 “participants underwent a speculum examination where 5% acetic acid was applied to the cervix using a cotton swab. VIA…“ for clarity it might be best to introduce the VIA abbreviation at this point

P16 “In brief, among the 2,969 women screened for HPV, 1,480 were positive and 1,269 had histopathology data for analysis“ the somewhat large discrepancy 1480 vs 1269 might warrant elaboration to exclude potential biases.

P16 “The health outcome is the number of histologically confirmed cases of CIN2+.“ Consider presenting the total number of identified CIN2+ cases of the 1269 histopathology assessed cases since this is the basis of later calculations. Possibly a number of true negative cases would also be informative.

P18-20 it might be neccessary to emphasize in Table 1,Table 2 and Table 3 that primary screening was done with hrHPV genotyping since this is not the most common screening method. Using another primary hrHPV test would make the triage by 8type restricted HR HPV genotyping another added cost

P20 since the discrepancy between finally examined cases and total HPV positive cases was not explained, it is not completely clear how the missing 211 cases affect calculation of true Cin2+ cases?

P20 copy paste errors “personnel (VIA Base: $followed by supplies“

P21 “As all strategies included the cost of hrHPV DNA testing..“ testing should be replaced by genotyping to avoid misleading the readers. This also applies to other places within the manuscript where the reader might be mislead that any hrHPV primary testing would have the same outcomes/costs

Reviewers' comments:

Reviewer's Responses to Questions

**Comments to the Author**

1. Is the manuscript technically sound, and do the data support the conclusions?

Reviewer #1: Yes

Reviewer #2: Partly

2. Has the statistical analysis been performed appropriately and rigorously?

Reviewer #1: N/A

Reviewer #2: Yes

3. Have the authors made all data underlying the findings in their manuscript fully available?

Reviewer #1: Yes

Reviewer #2: Yes

4. Is the manuscript presented in an intelligible fashion and written in standard English?

Reviewer #1: Yes

Reviewer #2: Yes

Reviewer #1: Authors Wynn et al present a costing study comparing three methods for HPV triage (8-type genotyping, VIA and colposcopy) in Botswana. hrHPV testing with 8-type genotyping is presented as the best strategy for triaging HPV positive women; a finding which is importantly consistent irrespective of HIV status. Overall, I found the collection and reporting of costs to be thorough and clear.

I have suggested minor revisions for this paper, and my suggested revisions are thus:

(1) The measure used in this analysis are not what most people consider to be "cost-effectiveness". ICERs are typically calculated as the incremental cost over the incremental benefit. In the field of cervical cancer prevention, "benefit" is typically defined as precancer or cancer prevented by screening. In your study, you have defined the benefit as true CIN detections, which is simply not the standard for this field (or the NICE guidelines). That is not to say that this measure of efficiency isn't important, however. I suggest re-branding this outcome as an "efficiency measure" rather than an ICER, and also removing the claim that this is a cost-effectiveness study from the title.

(2) While it is transparent in the discussion that only costs for primary screening and triaging are considered (rather than total screening pathway costs), it is my opinion that more discussion about the possible impact that this would have on the findings is warranted. Screening approaches with low specificity (such as screen-and-treat with HPV and hrHPV with 8-genotype triaging) are likely to appear less expensive (relative to how expensive they currently look compared to high-specificity approaches) once repeat HPV testing of triage-negative women (and any subsequent precancer treatment) is considered. This is likely to be even more significant for settings with high HIV prevalence, as women living with HIV are more likely to be carrying a persistent HPV infection than HIV-negative women.

(3) A minor comment but please check ref 19 in the introduction. The referenced paper is very old and doesn't seem to support your statement.

(4) Suggest including the country (Botswana) in the title.

Reviewer #2: It’s my pleasure to have the opportunity to review the manuscript. I have gone through the manuscript in detail and please see my comments below.

Abstract

1. The authors stated that the analysis was conducted among women with and without HIV. However, the results did not mention any stratification regarding the findings. It could be better to state upfront that results are consistent in women with and without HIV.

2. The results stated that “personnel and hrHPV test kit cartridge costs as significant drivers of overall costs.” What’s the reason to only emphasize the need to reduce the prices for HPV tests?

3. The difference between “long-term” vs. “short-term” is not well defined in the abstract.

Introduction

1. When stating that “hrHPV testing has the potential for rapid scale- up and increased access to screening through self-sampling, making it a highly attractive option for screening in sub-Saharan Africa”, have there been studies assessing the acceptance and performance of self-sampling in the region?

2. hrHPV test usually has a lower specificity compared to cytology.

Koliopoulos G, Nyaga VN, Santesso N, Bryant A, Martin‐Hirsch PP, Mustafa RA, Schünemann H, Paraskevaidis E, Arbyn M. Cytology versus HPV testing for cervical

3. Cytology triage is a common triage method for hrHPV-positive results. Could the authors expand on why the test “is not available at a population level”?

4. At the end of the introduction, please add that the analysis was conducted among both women with and without HIV.

Methods

1. Subtitles have different formats in terms of capitalizing the first letters. Please keep them consistent.

2. Please specify which 8 types of HPV were considered eligible to be treated in text and also cite to support the choice of the 8 HPV type to be eligible for treatment.

3. Is the cost estimated as per screening or per woman who tested hrHPV positive? Are these two equivalent?

4. The number of procedures observed to determine personnel utilization seems to vary a lot by the triage test (ranging from 6 to 20). Does it depend on the complexity of the procedure?

5. Does recording the number of supplies used during a procedure require “timing”? Why was the “time-and-motion observation” used to assess the number of supplies utilized?

6. Was the CHEERS only apply to the post-hoc analysis? If not, it would be better to state it upfront in the “Study design” section.

7. The post-hoc analysis seems to be a crucial portion as it reflects the “short-term” cost of the treatment. It seems more appropriate to include the “post-hoc” in the main analysis and justify the significance of including it in the study. Also, please define “short-term” earlier in the text.

8. The sensitivity and specificity of the three triage strategies seem to fit in the results section.

9. Colposcopy is usually considered the “gold-standard” diagnostic test for detecting CIN2+. It seems that in the current study, colposcopy performance was less ideal and resulted in a lower sensitivity than hrHPV triage, especially in women without HIV.

10. It would be better for the readers to understand the context if the authors included a flowchart to describe the screening scheme of the three triage strategies implemented in Botswana.

11. This is my main question/concern: in real-world settings, when using hrHPV triage, do all women who test positive receive treatment (and what type of treatment would that be), or is a colposcopy still needed to determine the treatment eligibility? If it is the latter, then it’s worrisome, given the colposcopy performance in the current study. In the current study, every hrHPV-positive woman had a histology sample collected to determine their “true health state.” However, in the real world, many true cases of CIN2+ would have been missed if there was “no visible lesion” and not every hrHPV positive was followed to collect a histology sample.

Results

1. Table 3: please specify what the numbers in the parenthesis indicate for the column “True CIN2+ Cases Detected”. Please also add the same range for the costs.

2. Please also add a column of false-positive results in Table 3 to help understand the specificity of the three triage methods.

3. Please revise the sentence: “For VIA and colposcopy triage, personnel (VIA Base: $followed by supplies were the biggest cost drivers.”

4. Please revise the sentence: “We also and found that genotyping became more costly ….”

5. Could the author please explain the potential reasons that the test sensitivity for VIA and colposcopy seemed to differ in women with and without HIV?

Discussion

1. Please include some comments on the differential performance of the VIA and colposcopy regarding patient’s HIV status.

2. Please discuss the implementation of the hrHPV triage in real-world settings (with or without a diagnostic colposcopy).

**Do you want your identity to be public for this peer review?** For information about this choice, including consent withdrawal, please see our Privacy Policy

Reviewer #1: No

Reviewer #2: No

---

## [Author Response · Author response to Decision Letter 1]

4 Apr 2025

Please see our response to reviewers document. For the editor, I could not find the financial disclosures that were inconsistent with the list of grants.

---

## [Decision Letter · Decision Letter 1]

7 May 2025

Dear Dr. Wynn,

**While the study undeniably adds interesting cost data and findings, one major concern was not addressed, which when dealing with patient oriented studies, is critical. The chosen triage strategy assumes immediate treatment and in this situation specificity of 30% (commented upon in the review process) cannot be ignored or remain unmentioned in the discussion section. The authors do state in the results section 'We found that genotyping became costlier than VIA when treatment costs exceeded $10 and costlier than colposcopy when they exceeded $36.'. This finding must be discussed in the context of significant overtreatment costs of the false positive cases even if other risks of overtreatment are ignored.**
** **
**Some minor formatting or typographical issues remain to be addressed.**

We look forward to receiving your revised manuscript.

Kind regards,

Ivan Sabol

Academic Editor

PLOS ONE

**Journal Requirements:**

**Additional Editor Comments:**

Page refers to the page number in the PDF document since manuscript page numbers are not shown.

p12 wrong reference and or reference format (superscript number 7) “(WLHIV) in Botswana.(6) 7”

p13 wrong reference and or reference format (superscript number 28) “Economic Evaluation Reporting Standards (CHEERS).(25)28”

p17 typo “The health outcomes have been previously reported(7)” no space

p18 “based on reviewer comments,” obsolete

p18 typo “positive cases of precancerin”

p20 wrong reference and or reference format (superscript number 8) “and 412 confirmed cases of CIN2+. 8”

p21 it is not necessary to list all 3 panels for the in text Figure 1 callout “(Figure 1. One-way sensitivity analysis of: 1) Primary hrHPV screen, which includes the genotyping triage strategy, 2) Additional cost of VIA as a triage screen, 3) Additional cost of colposcopy as a triage screen.)”

p21 typo “personnelfollowed”

p22 in text Figure 2 callout doesn’t need to contain the title

p24 typo missing fullstop “primary HPV result In this”

p29 Figure 1 title of panel 1 typo “hHPV”, x-axis title is cropped out “(genotyping is no…”

p30 Figure 2 y axix title has typo “fof”

Reviewers' comments:

Reviewer's Responses to Questions

**Comments to the Author**

Reviewer #2: (No Response)

2. Is the manuscript technically sound, and do the data support the conclusions?

Reviewer #2: Yes

3. Has the statistical analysis been performed appropriately and rigorously?

Reviewer #2: Yes

4. Have the authors made all data underlying the findings in their manuscript fully available?

Reviewer #2: No

5. Is the manuscript presented in an intelligible fashion and written in standard English?

Reviewer #2: Yes

**Reviewer #2: ** Thanks for inviting me to review the revision. The authors addressed most of my comments. However, my main concern was still not addressed.

One of my previous comments raised my question on why colposcopy, usually considered the “gold standard” for detecting lesions, was less ideal and had lower sensitivity than hrHPV triage. The authors mentioned the challenge of collecting endocervical curettage without a visible lesion, which is understandable. However, the high sensitivity of the hrHPV triage in the current study is conditioned on a universal biopsy collection regardless of whether a lesion is visible during a colposcopy. Although hrHPV genotyping triage has a high sensitivity, as it will identify more women with true disease, if the diagnostic procedure itself has low performance, we still face the dilemma of the next steps after a positive genotyping triage result. If the next step requires a diagnostic test (VIA or colposcopy) to ensure treatment, the performance of the tests is less ideal as many true cases will still be missed; if the next step is immediate treatment, we suffer from the low specificity (30% as mentioned in the manuscript) and overtreatment. The authors will need to address the limitations of the colposcopy (or other diagnostic procedure) performance. Otherwise, the findings from the current study and the benefit of the high sensitivity of the hrHPV genotyping triage may not be utilized in real-world settings.

**Do you want your identity to be public for this peer review?** For information about this choice, including consent withdrawal, please see our Privacy Policy

Reviewer #2: No

---

## [Author Response · Author response to Decision Letter 2]

17 Jun 2025

Dear Academic Editor and Reviewers,

Thank you for continuing to review our research article entitled, "Short-term costs and cost-efficiency of HPV triage strategies in a high HIV-prevalence setting: Evidence from Botswana.” Please see our responses to each comment below.

Minor

• p12 wrong reference and or reference format (superscript number 7) “(WLHIV) in Botswana.(6) 7” Fixed

• p13 wrong reference and or reference format (superscript number 28) “Economic Evaluation Reporting Standards (CHEERS).(25)28” Fixed

• p17 typo “The health outcomes have been previously reported(7)” no space Fixed

• p18 “based on reviewer comments,” obsolete Removed text

• p18 typo “positive cases of precancerin” Fixed

• p20 wrong reference and or reference format (superscript number 8) “and 412 confirmed cases of CIN2+. 8” Updated

• p21 it is not necessary to list all 3 panels for the in text Figure 1 callout “(Figure 1. One-way sensitivity analysis of: 1) Primary hrHPV screen, which includes the genotyping triage strategy, 2) Additional cost of VIA as a triage screen, 3) Additional cost of colposcopy as a triage screen.)” We removed the titles

• p21 typo “personnelfollowed” Fixed

• p22 in text Figure 2 callout doesn’t need to contain the title We removed the title

• p24 typo missing fullstop “primary HPV result In this” Fixed

• p29 Figure 1 title of panel 1 typo “hHPV”, x-axis title is cropped out “(genotyping is no…” Fixed

• p30 Figure 2 y axix title has typo “fof” Fixed

Editor Comments

While the study undeniably adds interesting cost data and findings, one major concern was not addressed, which when dealing with patient oriented studies, is critical. The chosen triage strategy assumes immediate treatment and in this situation specificity of 30% (commented upon in the review process) cannot be ignored or remain unmentioned in the discussion section. The authors do state in the results section 'We found that genotyping became costlier than VIA when treatment costs exceeded $10 and costlier than colposcopy when they exceeded $36.'. This finding must be discussed in the context of significant overtreatment costs of the false positive cases even if other risks of overtreatment are ignored. We agree and have added content on overtreatment costs in our response to Reviewer 2 below.

Reviewer 2

One of my previous comments raised my question on why colposcopy, usually considered the “gold standard” for detecting lesions, was less ideal and had lower sensitivity than hrHPV triage. The authors mentioned the challenge of collecting endocervical curettage without a visible lesion, which is understandable. However, the high sensitivity of the hrHPV triage in the current study is conditioned on a universal biopsy collection regardless of whether a lesion is visible during a colposcopy. Although hrHPV genotyping triage has a high sensitivity, as it will identify more women with true disease, if the diagnostic procedure itself has low performance, we still face the dilemma of the next steps after a positive genotyping triage result. If the next step requires a diagnostic test (VIA or colposcopy) to ensure treatment, the performance of the tests is less ideal as many true cases will still be missed; if the next step is immediate treatment, we suffer from the low specificity (30% as mentioned in the manuscript) and overtreatment. The authors will need to address the limitations of the colposcopy (or other diagnostic procedure) performance. Otherwise, the findings from the current study and the benefit of the high sensitivity of the hrHPV genotyping triage may not be utilized in real-world settings.

Thank you for highlighting the implications of test performance beyond initial screening and triage. We agree that the utility of hrHPV genotyping as a triage strategy hinges on the performance of subsequent diagnostic or treatment procedures, and that weak performance at these later stages can diminish the value of high initial sensitivity. To address this, we have added the following underlined content to paragraph five of the discussion:

While our study included the costs of primary hrHPV screening and three strategies for triaging women who were hrHPV positive, we did not collect costs associated with the subsequent treatment and rescreening cascade. Compared to VIA and colposcopy triage, hrHPV genotyping more accurately narrows the pool of people who need further evaluation; however, the impact of this strategy is dependent on the subsequent diagnostic and/or treatment procedures. For example, if the diagnostic procedure is VIA, many true cases will still be missed. VIA and colposcopy have limitations, particularly in detecting endocervical lesions. Women with HIV may have increased inflammation and non-HPV-related cervicitis, making visible lesions more common and potentially larger, leading to higher VIA positivity rates.(31-33) Further, if the next step is immediate treatment (e.g. LEEP or thermal ablation), overtreatment will occur. This weakness is partially demonstrated by our post-hoc analysis that found hrHPV genotyping became costlier than VIA and colposcopy at relatively low treatment costs due to higher sensitivity and lower specificity. In long-term mathematical modeling studies, preventing the costs of cancer care can balance out the costs of overtreatment.(30) However, the harms caused by overtreatment are not well measured, there is no widely accepted threshold on the maximum number of cases of overtreatment that are acceptable in order to prevent an additional case of cervical cancer, and more research is needed.(34) In practice, the next steps following an 8-type hrHPV genotype restriction would vary by context. For example, higher-resource settings may opt for routine biopsies to determine treatment needs, while others may rely on VIA These findings highlight the need for tailored triage strategies that consider both resource availability and patient-specific factors such as HIV status.

---

## [Decision Letter · Decision Letter 2]

8 Jul 2025

Short-term costs and cost-efficiency of HPV triage strategies in a high HIV-prevalence setting: Evidence from Botswana.

PONE-D-24-33643R2

Dear Dr. Wynn,

We’re pleased to inform you that your manuscript has been judged scientifically suitable for publication and will be formally accepted for publication once it meets all outstanding technical requirements.

Kind regards,

Ivan Sabol

Academic Editor

PLOS ONE

Additional Editor Comments (optional):

Reviewers' comments:

Reviewer's Responses to Questions

**Comments to the Author**

Reviewer #2: All comments have been addressed

2. Is the manuscript technically sound, and do the data support the conclusions?

Reviewer #2: Yes

3. Has the statistical analysis been performed appropriately and rigorously?

Reviewer #2: Yes

4. Have the authors made all data underlying the findings in their manuscript fully available?

Reviewer #2: Yes

5. Is the manuscript presented in an intelligible fashion and written in standard English?

Reviewer #2: Yes

Reviewer #2: Thank you for the opportunity to review this manuscript. The authors have addressed all my concerns in the revision.

**Do you want your identity to be public for this peer review?** For information about this choice, including consent withdrawal, please see our Privacy Policy

Reviewer #2: No

---

## [Editor Report · Acceptance letter]

PONE-D-24-33643R2

PLOS ONE

Dear Dr. Wynn,

I'm pleased to inform you that your manuscript has been deemed suitable for publication in PLOS ONE. Congratulations! Your manuscript is now being handed over to our production team.

Kind regards,

on behalf of

Dr. Ivan Sabol

Academic Editor

PLOS ONE